# Genome-Wide Transcriptome Analysis Reveals That Upregulated Expression of *Aux/IAA* Genes Is Associated with Defective Leaf Growth of the *slf* Mutant in Eggplant

**Wenchao Du [1], Yang Lu [2], Shuangxia Luo [1], Ping Yu [1], Jiajia Shen [1], Xing Wang [1], Shuxin Xuan [1], Yanhua Wang [1], Jianjun Zhao [1], Na Li [1,*], Xueping Chen [1] and Shuxing Shen [1]**

[1]  Key Laboratory for Vegetable Germplasm Enhancement and Utilization of Hebei, Collaborative Innovation Center of Vegetable Industry in Hebei, College of Horticulture, Hebei Agricultural University, Baoding 071000, China
[2]  Hebei University R&D Center for Sericulture and Specialty Enabling Technologies, Institute of Sericulture, Chengde Medical University, Chengde 067000, China
*   Correspondence: yyln@hebau.edu.cn

**Abstract:** Leaf size is a crucial trait in eggplant breeding, as it influences photosynthesis, plant biomass and management. However, little is known about the molecular mechanism regulating leaf size in eggplant. This study reports a small leaf mutant (*slf*) generated with the mutagen ethyl methane sulfonate (EMS). The *slf* mutant showed restricted cell proliferation and an increased content of auxin. Transcriptome analysis revealed that several genes involved in auxin signaling are upregulated in *slf*. Exogenous application of auxinole, an auxin antagonist of TIR1/AFB receptors, repressed the expression of these genes and restored leaf growth of *slf*, suggesting that the small leaf size of *slf* is likely associated with auxin signaling. This study provides essential clues to unveil the molecular mechanism of leaf size regulation in eggplant.

**Keywords:** eggplant; leaf size; Aux/IAA; transcriptome analysis; auxinole

## 1. Introduction

Eggplant (*Solanum melongena* L.), also known as aubergine and brinjal, is one of the most economically important vegetable crops of the Solanaceae. It is widely produced in India and China [1–3] and has rich genetic diversity [4]. Leaves play crucial roles in plant fitness and stress responses [5–8] and are the primary site of photosynthesis. Their size affects photosynthetic capacity and causes variability in sugar contents and yield [9,10]. In eggplant, leaf size has a direct impact on plant growth and yield [11,12], compromising, in many cases, food security in poor areas [11].

Leaf size is determined by the complex coordination of cell proliferation and expansion [13]. These two different processes are strictly controlled by various integrated signals from the intrinsic genetic network and the growth environment [14], among which many proteins, such as CDC27a, GRFs, CYCD3, EXP10 and EBP1, are involved in positive regulation [15–19], while other proteins, such as DELLA, ARF2, DA1 and TCP4, are involved in negative regulation [20–23]. Among these, many factors, such as DELLA and ARFs, are important components of phytohormone transduction, showing that phytohormones are essential in controlling leaf size. However, little is known about the mechanism of leaf size regulation in eggplant.

Auxin is one of the most critical phytohormones in plants [24] and plays a key role in leaf size regulation [12,25,26]. Auxin regulates the transcription of auxin-responsive genes through the action of several key components, including Transport Inhibitor Response1/Auxin signaling F-Box proteins (TIR1/AFBs), Auxin/indole-3-acetic acid family proteins (Aux/IAAs) and Auxi Response Factors (ARFs) [27]. At lower auxin concentrations, TIR1/AFBs interact with Aux/IAA proteins to form an auxin coreceptor that

bonds with ARFs. At higher auxin concentrations, IAA or auxin-like substances enhance the interaction of TIR1/AFBs with Aux/IAA, promote ubiquitination and degradation of Aux/IAA [28–30] and then release ARFs [31,32], which activate the expression of auxin response genes [33–35].

TIR1/AFBs were demonstrated to be an auxin coreceptor and a controller of leaf development in *Arabidopsis* [36,37]. *Aux/IAAs*, auxin-responsive genes, can be induced rapidly by auxin and affect leaf morphogenesis, growth and development in many plants, including tomato [38–40], *Arabidopsis* [41] and tobacco [42], mediated by changes in transcript levels [43,44]. ARFs, which function as transcriptional activators/repressors, can confer changes in leaf size in plants by modulating the transcription of downstream genes (*Aux/IAAs*) in auxin signaling pathways [31,45,46]. In addition, the Auxin uptake carrier (AUX1) is also an important component in the auxin signaling pathway, as it was found to be a key protein involved in IAA uptake [47].

In this study, we characterized a leaf size mutant *slf* isolated from an EMS mutagenesis library of the inbred line '14–345'. The mutant *slf* had smaller leaves than the wild type. RNA-seq analysis revealed that the transcription of Aux/IAA genes was upregulated in the *slf* mutant. Auxin antagonist auxinole treatment largely restored leaf growth of the *slf* mutant.

## 2. Materials and Methods

### 2.1. Plant Materials and Growth Conditions

The *slf* mutant was isolated from the $M_2$ population of EMS mutagenized eggplant inbred line '14–345'. Three thousand seeds of '14–345' were mutagenized with 0.8% EMS to produce the $M_1$ population, and individuals from the $M_1$ population were self-pollinated to obtain $M_2$ seeds. Ten seeds obtained from each surviving $M_1$ plant were sown as one $M_2$ line to produce the $M_2$ population. Four hundred $M_2$ mutant lines were sown and one mutant with small leaf size was isolated and named *slf*. The *slf* mutant was self-pollinated to obtain M3 seeds. The *slf* mutant and WT seedlings were grown at Hebei Agricultural University in the spring of 2022 in a greenhouse at 28 °C with a 16/8 h light/dark photoperiod.

### 2.2. Leaf Size Measurement

The third leaf sizes of WT and slf were measured at the sixth-true-leaf stage using an A3 scanner. Fresh leaves were collected and scanned to obtain a picture. The surface area of each leaf was determined using a scanner (Type: Uniscan M1 Plus; Unis, Beijing, China) to obtain data. Then leaf area was analyzed by Image J ecosystem [48]. Leaves from six individual plants of WT and slf were measured. Three biological replicates were measured, each with two plants per genotype.

### 2.3. Histological Analysis

The mature leaf at the seventh internode of the plant was taken at 80 days after transplantation and was analyzed by differential interference contrast microscopy (DICM). Briefly, leaf samples from a similar area in mature leaves of WT and *slf* were collected and then kept in 10% chloral hydrate for 36 h. Fifty-micrometer sections were prepared (Olympus DP71). Three sections of one biological sample and approximately 100 cells were observed in our study.

### 2.4. Measuring the Main Hormone Content

Fresh leaf samples were taken from similar positions at the fourth-leaf stage, fifth-leaf stage and seventh-leaf stage, and approximately 0.5 g of each sample was used to determine the contents of abscisic acid (ABA), zeatin riboside (ZR), indole-3-acetic acid (IAA), brassinosteroid (BR) and gibberellin ($GA_3$). Hormone measurements were performed by using the enzyme-linked immunosorbent assay (ELISA) described by Popova et al. [49].

## 2.5. RNA-seq Analysis

Young leaves of WT and *slf* plants at the first-leaf stage 20 days after transplant and at fourth-leaf stages 40 days after transplant were collected. These samples were stored quickly in liquid nitrogen and then stored at −80 °C. Total RNA was extracted using an EASTEP Super Total RNA Kit (Promega, Shanghai, China) according to the manufacturer's instructions.

Three biological replicates were designed for each genotype. The RNA quality analysis, cDNA library preparation and sequence analysis were conducted by the Novogene Technology Company, Beijing, China. Quality of RNA was checked by determining the RNA integrity and concentration using RNase-free 1% agarose gel electrophoresis, RNA Assay Kit in Qubit2.0 Flurometer (Life Technologies, Carlsbad, CA, USA) and RNA Nano6000 Assay Kit of the Bioanalyzer 2100 system (Agilent Technologies, Palo Alto, CA, USA). Sequencing libraries were generated using the NEBNext UltraTM RNA Library Prep Kit for Illumina (NEB, Ipswich, MA, USA) following the manufacturer's recommendations. Index codes were added to attribute sequences to each sample [50,51]. The gene expression level, differential gene expression analysis and Gene Ontology (GO) enrichment were performed by using the feature Counts v1.5.0-p3, the DESeq2 R package (1.16.1) (Novogene Technology Company, Beijing, China) and the cluster Profiler R package [52–54].

## 2.6. Validation of Selected DEGs Using Real-Time Quantitative Reverse Transcription PCR (qRT-PCR)

Four μm of total RNA was reverse transcribed using a One-step gDNA Removal and cDNA Synthesis SuperMix kit (EasyScript@ AE311) according to the manufacturer's instructions. Tubulin gamma (Smechr0302615) was used for normalization [55], and the primer sequences for all genes analysed are listed in Table S1. qRT-PCR was performed with ChamQ Universal SYBR qPCR Master Mix. This reaction mixture was 20 μL and run in a BIO-RAD CFX96 TOUCH Real-time qPCR detection system (Bio-Rad, Hercules, CA, USA), with three biological replicates for each type of sample. The PCR program used was as follows: each reaction included 2 μL of 1:10 diluted cDNA, 10 μL SYBR Green, 0.5 μL forward primer, 0.5 μL reverse primer and 8.0 μL double-distilled (dd)$H_2O$. The PCR reaction was conducted as follows: initial activation at 95 °C for 3 min, followed by 40 cycles of 95 °C for 10 s, 57 °C for 30 s. The $2^{-\Delta\Delta Ct}$ method was used to analyze the relative expression levels of each gene [56]. The RNA samples used for RNA-seq were also used for qRT-PCR analysis.

## 2.7. Exogenous Auxinole Treatments

From the first-leaf stage, the second leaf of WT and *slf* plants were sprayed with 20 μm/L auxinole, an Aux/IAA expression inhibitor [57], which was dissolved in 10 μm/L DMSO. The treatments were conducted with one application every 3 days for 20 days. The plants treated with 10 μm/L DMSO were used as controls. The second-leaf sample for gene expression testing and investigation of leaf size was collected six hours after the last application of auxinole and DMSO. Three biological replicates were performed to evaluate the leaf size, each with five plants per genotype.

## 2.8. Statistical Analysis

For statistical analysis, Student's *t*-test was used [58]. All data represent the mean ± SD of at least three replicates. Asterisks denote significant differences (* $p < 0.05$; ** $p < 0.01$) as determined by Student's *t*-tests.

## 3. Results

### 3.1. Phenotypic Comparison of the WT and Slf Mutant

The *slf* plant showed a small leaf size phenotype from the seedling stage compared to that of the WT (Figure 1A,B). To further characterize the *slf* mutant, we compared the WT and *slf* mutant leaf sizes at mature stages 80 days after transplantation. The palisade layer

cell area of leaves from both the WT and *slf* was analyzed at the mature stage; the cell area of *slf* was 100% larger than that of WT, indicating that cell proliferation inhibition is key for leaf size changes in *slf* (Figure 1C,D,F). Moreover, the leaf size of the *slf* mutant was continuously smaller than that of the WT, significantly reduced to approximately 25.4% in *slf* (Figure 1E). The fruit size is approximately 10% smaller in *slf* than those in WT; however, more flowers in *slf* and good management keep a similar yield between slf and WT.

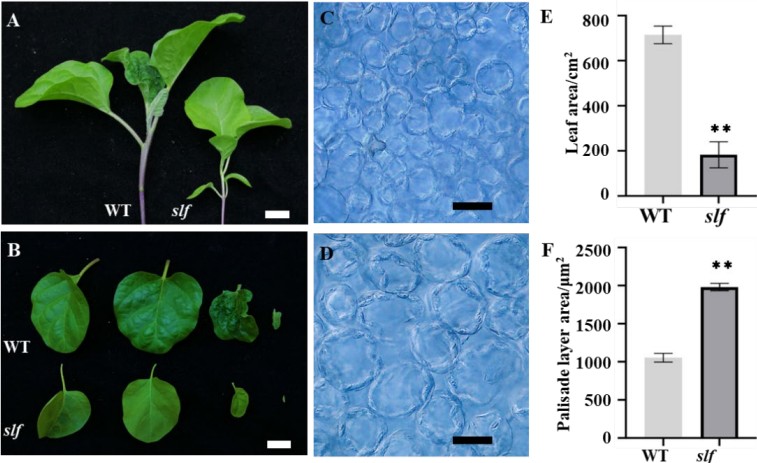

**Figure 1.** Phenotypic analysis of WT (14–345) and *slf*. (**A**) WT and *slf* plants at the seedling stage, (**B**) leaf size of the WT and *slf* at the seedling stage; white scale bar = 1 cm. Mature leaf cell area of palisade layer of WT (**C**) and *slf* (**D**), scale bars = 50 μm; analysis of mature leaf area of WT and *slf* (**E**) and palisade layer call area of WT and *slf* (**F**). ** indicates significant differences compared with the WT (Student's *t*-test, $p < 0.01$). Values are the mean $\pm$ SE of three biological replicates.

*3.2. Analysis of the Contents of ABA, IAA, BR, GA3 and ZR*

We tested the contents of ABA, IAA, BR, GA3 and ZR in *slf* and WT leaves. A significantly higher content of IAA was observed in *slf* at different growth stages from the fourth-leaf stage to the seventh-leaf stage, whereas the contents of ABA, BR, GA3 and ZR differed between *slf* and WT at different growth stages (Figure 2). These results indicated that the small leaf size in *slf* might be related to altered auxin homeostasis or auxin signaling.

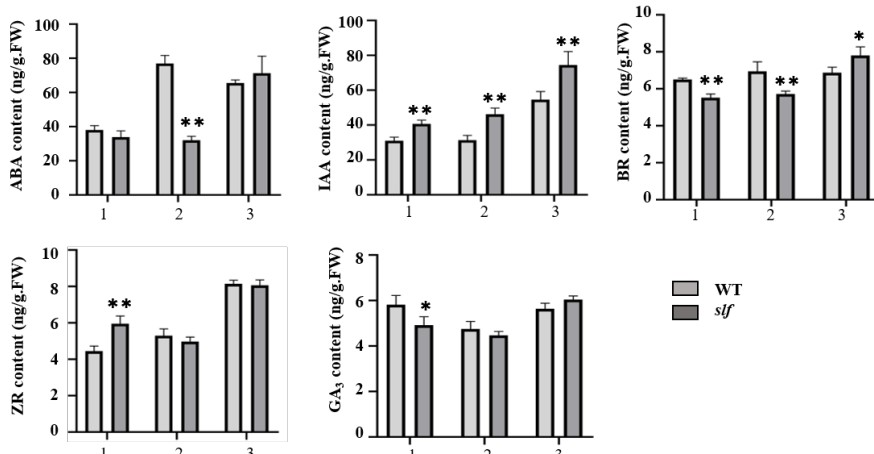

**Figure 2.** The contents of ABA, IAA, ZR, BR and GA3 in WT (14–345) and *slf* leaves at the fourth-leaf stage (1), fifth-leaf stage (2) and seventh-leaf stage (3). Asterisks denote significant differences (* $p < 0.05$; ** $p < 0.01$) as determined by Student's *t*-tests.

### 3.3. Transcriptome Profiles Showed That Genes Involved in Auxin Signalling Are Enhanced in Slf

To further characterize the genes involved in the small leaf size of the *slf* mutant, we performed RNA-seq experiments using total RNA isolated from young leaves of the *slf* mutant and WT. High-throughput RNA-seq generated 40.22–51.14 million raw reads for each sample and 2.68 hundred million raw reads for all six libraries (Supplementary Materials Table S2). After the original data were filtered and adapter sequences were removed, 39.66–50.61 million clean reads were mapped to the eggplant genome (Supplementary Materials Table S2).

Transcriptome analysis identified totally 1207 differentially expressed genes (DEGs) in first-stage leaves of *slf* and WT (Figure 3A), 851 of which were upregulated and 356 were downregulated in *slf*.

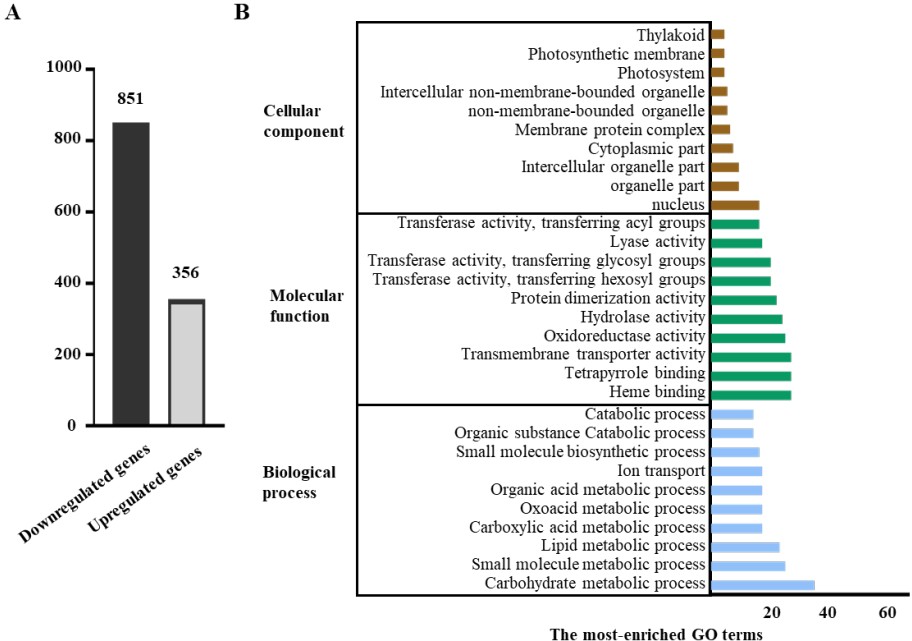

**Figure 3.** Differentially expressed genes in WT (14–345) and *slf* leaves. (**A**) Total numbers of differentially expressed genes of WT and *slf* leaves; (**B**) GO classification of DEGs in WT and *slf* leaves. The X-axis represents the number of genes annotated into the GO terms, and the Y-axis represents the functional classification.

To understand the potential function of the DEGs, we performed GO classification analysis of DEGs and obtained the top ten GO terms in biological process, cellular component and molecular function. GO: 0016052 (carbohydrate catabolic process) was the most highly enriched term among the biological processes. Of the ten dominant terms in molecular function were GO: 0020037 (heme binding), GO: 0046906 (tetrapyrrole binding) and GO: 0022857 (transmembrane transporter activity). GO: 0005634 (nucleus) was highly enriched in cellular components (Figure 3B).

Kyoto Encyclopedia of Genes and Genomes (KEGG) term enrichment analyses were performed in upregulated or downregulated genes of two comparison sets (Figure 4). Remarkably, the KEGG term "plant hormone signal transduction" was enriched in upregulated genes in *slf* compared to WT (Figure 4A). In addition, upregulated genes involved in the "carbon metabolism", "starch and sucrose metabolism", "mRNA surveillance pathway" and "plant pathogen interaction" terms were enriched in *slf* compared to WT (Figure 4A). Among the downregulated DEGs, genes related to "carbon metabolism" were significantly enriched in *slf* in comparison to WT at the first-leaf stage (Figure 4B), while genes related to "carbon fixation in photosynthetic organisms" and "glyoxylate and dicarboxylate metabolism" were also significantly enriched in the *slf* in comparison to WT group (Figure 4B). However, the downregulated DEGs were related to the essential metabolic

processes of plants. Thus, transcriptional activation essentially targeted the mechanism controlling leaf development in eggplant. DEG analysis further showed that six upregulated genes were enriched in the plant hormone transduction pathway (Figure 4A), five were enriched in auxin signaling (Figure 5A,B) and three were related to Aux/IAA (Figure 5). The other upregulated genes *Smechr0100111* and *Smechr0402457* in *slf* encode AUX1 and ARF5, respectively. These studies indicated that auxin signaling is enhanced and plays a key role in regulating leaf size in *slf*.

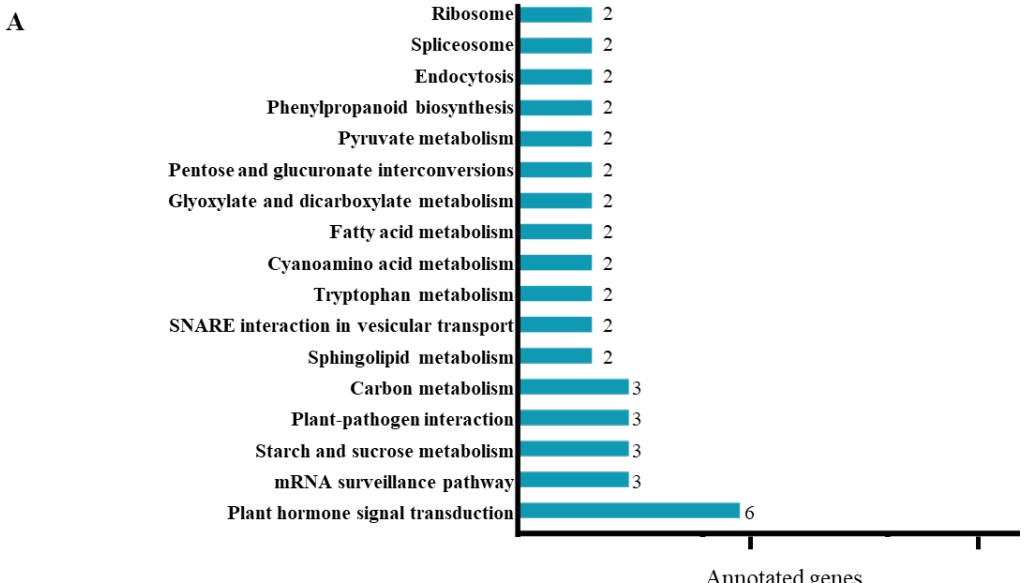

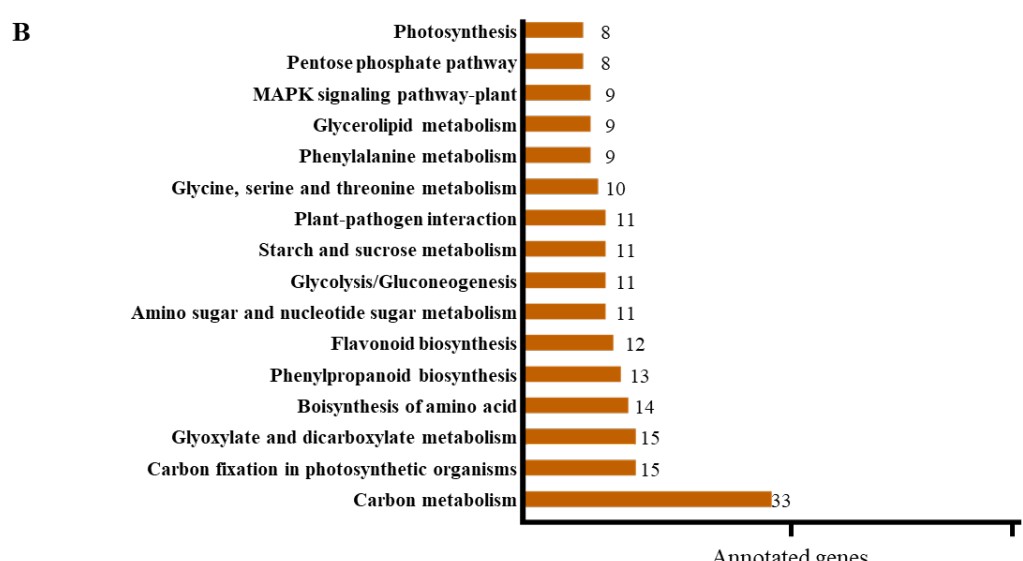

**Figure 4.** Statistical analysis of annotated unique genes in KEGG pathways in WT (14–345) vs. *slf*. (**A**) Upregulated genes in KEGG pathways in WT vs. *slf*. (**B**) Genes downregulated in KEGG pathways in WT vs. *slf*.

To validate the RNA-seq data, the expression levels of DEGs were measured using quantitative real-time PCR (qRT-PCR). A total of eight genes related to auxin signaling and the pathogen interaction pathway were selected (Figure 6A and Supplementary Materials Table S1). The overall correlation coefficient of a linear regression analysis was 0.8796 (Figure 6B).

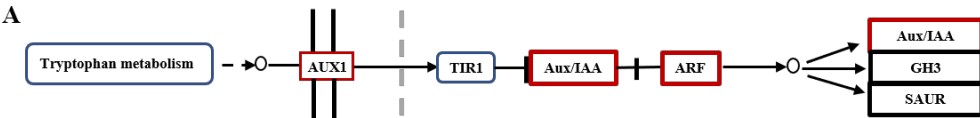

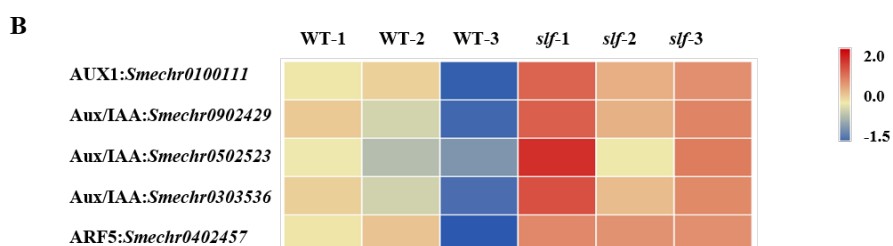

**Figure 5.** Differentially expressed genes (DEGs) in the auxin signaling pathway. (**A**) Auxin signaling pathway regulates the expression of auxin-responsive genes through the F-box protein TIR1 and cooperative action of Aux/IAA transcriptional repressors and ARFs. Red rectangles represent genes that were significantly upregulated in *slf* vs. WT (**B**) Expression levels of DEGs in the auxin signaling pathway in the WT and *slf* samples. Values are the mean ± SE of three biological replicates.

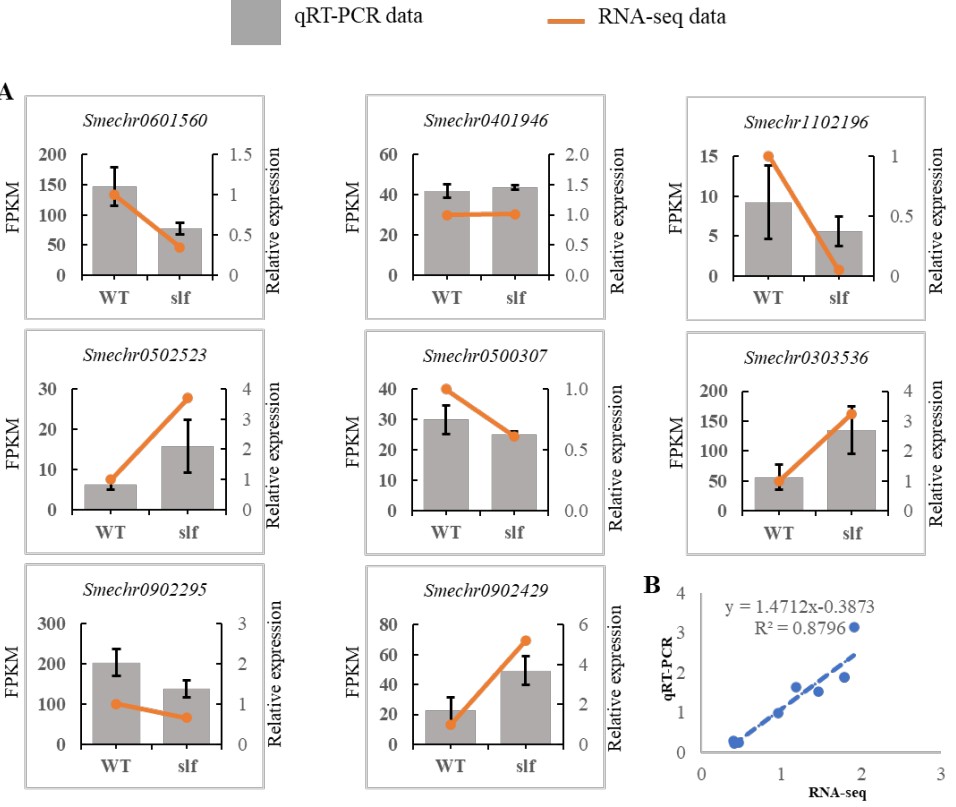

**Figure 6.** qRT-PCR verified the results of RNA-seq. The column diagram and line chart represented the data from qRT-PCR and RNA-seq, respectively. (**A**) The gene expression levels of qRT-PCR were normalized using the relative expression level of the internal control, tubulin gramma and WT (14–345) expression levels were normalized to 1. The data presented are the means from three biological replicates. (**B**) Pearson's correlation of gene expression ratios (*slf*/WT) between RNA-seq and qRT-PCR. The correlation of the fold change was analyzed by RNA-seq (x-axis) with qRT-PCR (y-axis) data.

*3.4. The Effect of Exogenous Auxinole on the Leaf Size of WT and slf*

To validate whether the upregulated *Aux/IAA* genes play key roles in the mechanism of small leaf formation in *slf,* leaf size and the expression level of *SmAux/IAA* under auxinole treatment were studied.

Under control treatment, *slf* shows a small leaf compared to WT; however, the leaf size of the *slf* mutant was significantly increased under auxinole treatment (Figure 7A,B). We then tested the expression of *Smechr0902429*, *Smechr0303536* and *Smechr0502523*, which have been shown to be upregulated in *slf*, as shown in transcriptome analyses. These three genes were significantly downregulated in both WT and *slf* under auxinole treatment compared to the control (Figure 7C). These data suggested that enhanced auxin signaling and upregulation of the Aux/IAA genes are responsible for the small leaf phenotype of *slf*. Interestingly, other *SmAux/IAA* genes, such as *Smechr0601560* and *Smechr0101209*, were changed little in *slf* but decreased significantly in WT under auxinole treatment compared to the control (Supplementary Materials Figure S1), and the leaf size of WT was slightly decreased compared to that of the control (Figure 7A,B).

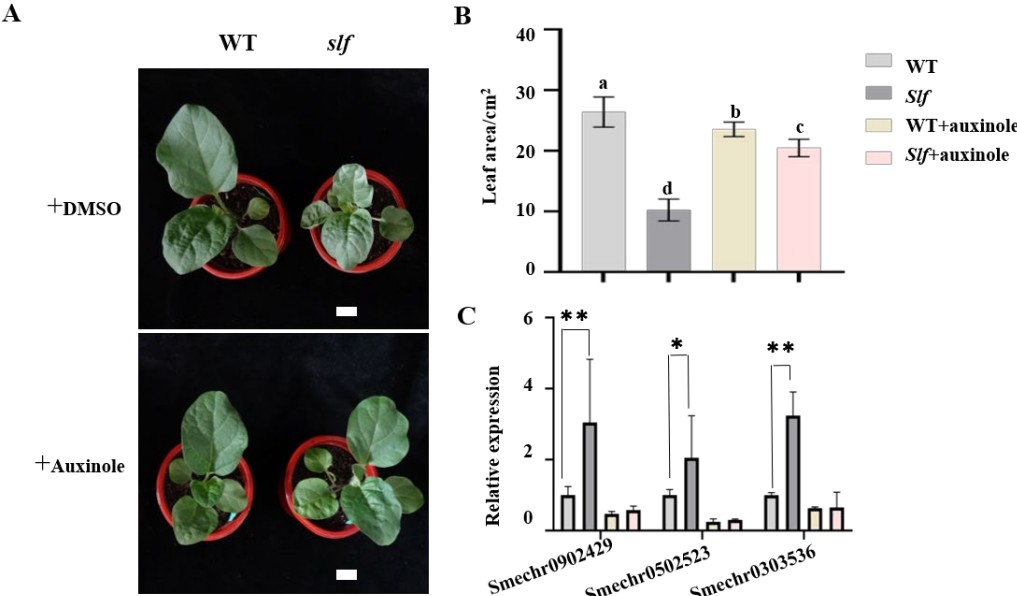

**Figure 7.** Effect of auxinole on leaf growth. (**A**) Leaf size under treatment with 10 μm/L DMSO or 20 μm/L auxinole. Bar = 1 cm. (**B**) Analysis of the leaf area 20 days after auxinole treatment. Values are the mean ± SE of three biological replicates. Different lowercase letters indicate significant differences ($p < 0.05$) among treatments for each column or parameter. (**C**) Expression levels of some auxin response genes treated with auxinole. The asterisks show significant differences (*t*-test; *, $p < 0.05$; **, $p < 0.01$) between the plants treated with control and auxinole.

## 4. Discussion

Leaf size has considerable economic relevance in crops [59], as it directly influences yield [60,61] or stress tolerance [62]. Therefore, understanding the mechanism of leaf development is essential to improve crop management. This study identifies a small leaf size mutant, *slf,* and shows that auxin signaling plays an important role in leaf size regulation in eggplant.

To further characterize the genes involved in the small leaf size of the *slf* mutant, the RNA-seq experiments were performed. GO annotation and GO enrichment analysis were conducted to understand the probable functions of the DEGs and the difference of DEGs between the WT and *slf*. The unique DEGs were enriched in the carbohydrate metabolic process (Figure 3B) which is highly associated with leaf development [63].

KEGG analysis showed plant hormone signal transduction was the representative pathway of DEGs in *slf* vs. WT (Figure 4). Shwartz found that hormone signal transduction

was highlighted during leaf development [64]. Interestingly, five out of six DEGs involved in auxin signaling pathway were upregulated in *slf* vs. WT. The other DEG related to plant hormone pathway was related to cytokinin. These results indicated that auxin signal was dominant during leaf development in *slf*.

The DEGs in the auxin signaling pathway were related to *AUX1* (*Smechr0100111*), *ARF5* (*Smechr0402457*) and three *Aux/IAA* genes.

AUX1 is an IAA transmembrane transporter [65], and it acts as a regulator in cell elongation but not in cell proliferation in *Arabidopsis* [66]. In this study, the *slf* mutant had small leaves but increased leaf cell size (Figure 1C,D,F), indicating that cell proliferation was inhibited and AUX1 is not the key factor in controlling leaf size of *slf* in eggplant. However, the detailed mechanism of Aux1 in the leaf development needs further exploration.

ARFs function as transcription factors in modulating the expression of downstream genes such as *Aux/IAA* [31]. It was also reported that ARF5 is involved in leaf vascular pattern regulation [44], and mutant analysis showed that ARF5 also acts in the process of leaf vein development; what is more, the *arf3 arf5* double mutant does not form leaves in *Arabidopsis* [67]. It was also found that ARF5 plays a key role in leaf development in grapevine [68]. These reports suggested that ARF5 is an important transcription factor related to leaf development. However, no research currently shows whether or not ARF5 is directly involved in the regulation of leaf size. Aux/IAAs were reported to be involved in leaf growth or leaf morphogenesis [39,42]. Additionally, considering that *Aux/IAA* is activated by and functions downstream of ARFs, *Aux/IAAs* were the focus of this study to elucidate the mechanism of leaf size of *slf* in eggplant.

To further characterize the function of upregulated expression of *Aux/IAAs* of *slf* in leaf size controlling, auxinole was used to repress the expression level of *Aux/IAA*. Auxinole is an auxin antagonist of TIR1/AFB receptors, and molecular analysis indicates that the auxinole strongly interacts with TIR1/AFB family members, which can repress the expression of earlier auxin responsive genes such as *Aux/IAAs* in *Arabidopsis* or other plants such as the moss *Physcomitrella patens* [57,65,69]. Exogenous auxinole inhibited the transcriptional expression of *Smechr0902429, Smechr0303536* and *Smechr0502523* and recovered the leaf size in *slf* mutants. Interestingly, the result also shows that the leaf size of WT was slightly inhibited compared to the control. Further investigation showed that the transcript levels of some *SmAux/IAAs*, such as *Smechr0601560* and *Smechr0101209*, were downregulated in WT under auxinole treatment, but in *slf*, the expression showed no significant difference compared to the control (Figure S2). We hypothesized that the function of auxinole may be partially inhibited in the *slf* mutant compared to WT, leading to uncontrolled transcriptional activation of the transcript level of some *Aux/IAAs*. The break-in transcriptional homeostasis of *Aux/IAA*, such as *Smechr0601560* and *Smechr0101209,* may contribute to the decrease in leaf size in WT plants. These results indicated that transcriptional changes in *Smechr0902429, Smechr0303536* and *Smechr0502523* play key roles in regulating leaf growth. In addition, the function of other *Aux/IAA* family members, such as *Smechr0601560* and *Smechr0101209*, deserve deeper study in eggplant.

The endogenous IAA content is continuously higher in *slf* than in WT, a conclusion that coincides with other research [70]. This should be a result that is regulated by other factors. Although reports showed that IAA content was tightly connected with the transcript level of *Aux/IAAs* or auxin signaling [31,35], more studies are needed to elucidate the reason for the continuously higher level of IAA contents in *slf*.

Further detailing the important role of auxin was addressed between the auxin signaling pathway and some enriched GO or KEGG terms. In *Prunus sibirica*, transcriptomic data and gene co-expression network analysis characterized that the ARF-related genes involved in carbohydrate metabolic process and were mediated by ARFs such as PsARF3 or PsARF5 [71]. Other enriched GO terms in our result such as transmembrane transporter activity were also reported to relate to auxin signaling [72,73]. KEGG analysis showed that one DEG (*Smechr0101117*) which encodes a histidine phosphotransfer protein (AHP) was found upregulated in *slf* and enriched in the cytokinin signaling pathway; research

showed that the transcription of *Aux/IAA* initiated by type-B response regulators (ARR) and this process was mediated by phosphorylated AHP [74]. In addition, other enriched KEGG pathways in our result such as phenylpropanoid biosynthesis or the starch and sucrose pathway were also reported to be correlated with the auxin signaling pathway. Jasmina Kurepa found that auxin sensitivity is controlled and fine-tuned by a cinnamate-4-hydroxylase step in early phenylpropanoid biosynthesis in *Arabidopsis* [75]. In tomato, overexpression of SlARF10 improved the accumulation of starch and sucrose in fruit, while SlARF10-RNAi lines showed decreased accumulation of starch and sucrose [76]. These results further imply the key role of the auxin signaling pathway in the leaf size regulation in eggplant.

### 5. Conclusions

In this study, we characterize an EMS mutant (*slf*) with an unusually small leaf size by EMS mutagenesis. Transcriptome analyses revealed that 1207 genes were differentially expressed in the first-leaf stage of the *slf* mutant compared to the WT. Hormone concentration measurements and KEGG pathway analyses indicated that the auxin signaling pathway was significantly affected in the *slf* mutant. Three Aux/IAA-related genes from the DEG list, *Smechr0902429*, *Smechr0303536* and *Smechr0502523*, have much higher expression in the *slf* mutant than in the wild type. Auxin antagonist auxinole treatment repressed the transcription of *Smechr0902429*, *Smechr0303536* and *Smechr0502523* and largely restored the leaf growth of the *slf* mutant, indicating that the Aux/IAA transcript level plays an important role in *slf*. Our findings offer a germplasm resource to study the mechanistic connection between auxin and leaf growth of eggplant, which would help facilitate eggplant breeding with ideal leaf sizes.

**Supplementary Materials:** The following supporting information can be downloaded at: https://www.mdpi.com/article/10.3390/agronomy12112647/s1, Table S1: Primers for qRT-PCR analysis; Table S2: Summary of read numbers in WT and *slf*; Figure S1: Expression levels of some other auxin response genes after treatment with auxinole. The asterisks show significant differences (*t*-test; * $p < 0.05$, ** $p < 0.01$) between plants treated with control and auxinole.

**Author Contributions:** Conceptualization, N.L., X.C. and S.S.; Writing—review and editing, W.D.; Validation, Y.L., S.L., P.Y. and J.S.; Methodology, Y.L., X.W., S.X., Y.W. and J.Z.; Funding acquisition, X.C. and S.S.; Project administration, N.L., X.C. and S.S. All authors have read and agreed to the published version of the manuscript.

**Funding:** This research was funded by the National Natural Science Foundation of China (grant no. 32172567), the Vegetable Innovation Team Project of Hebei Modern Agricultural Industrial Technology System (grant no. HBCT2018030203), the Key Research & Development Project of Hebei Province (grant no. 21326309D), The Innovation Ability Training Project for Graduate Student of Hebei Province (grant no. CXZZBS2018114), and the grant from 'Giant Plan' of Hebei Province.

**Institutional Review Board Statement:** Not applicable.

**Informed Consent Statement:** Not applicable.

**Data Availability Statement:** Not applicable.

**Acknowledgments:** We thank Wei Ma and Lisong Ma for their helps in preparation of this manuscript.

**Conflicts of Interest:** The authors declare no competing interests.

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
