# Peer review of "Genome-Wide Transcriptome Analysis Reveals That Upregulated Expression of Aux/IAA Genes Is Associated with Defective Leaf Growth of the slf Mutant in Eggplant"

_agronomy, doi:10.3390/agronomy12112647_

Round 1
Reviewer 1 Report
The authors have investigated the molecular mechanism regulating leaf size in eggplant based on a small leaf mutant (slf) using transcriptome analysis with several genes involved in auxin signalling identified as upregulated in slf. The authors further concluded that the small leaf size of slf is probably associated with auxin signalling. I believe that the authors have provided sufficient background, detailed explanations of well-established methodologies, presented the results with appropriate tables and figures, and concluded appropriately based on available data. The overall presentation of this manuscript is well. However, I have a couple of major and several minor concerns listed below for the authors to consider if a revision is requested by the editor.
Line 30, remove “family”
Line 73, the authors claimed that the leaf size mutant slf was isolated by EMS from ’14-345’…however, no detailed experiments of this isolation were provided. I believe it is important to provide some detailed explanations of how this mutant was isolated.
Line 77, the leaf size measurement is not clearly described. This needs more detailed explanations. A figure may help demonstrate where and how the leaf measurement is done. This figure would be also helpful for other experiments as well.
Line 109, use “4 µm” but not “Four micrograms”
Line 115: the PCR protocol:
Line 143, the species name of eggplant needs to be given in the figure caption; and this is the same problem for other figures as well. The labels of Y-axis of parts E and F need to be moved closer to the frame, it is too spacious now…
Line 204, the labels of % on the X-axis are confusing…
Line 217, parts A and B of figure 6 are not explained in the figure caption.
Discussion: I believe that the Discussion needs substantial improvement because a large amount of the results is not discussed in-depth at all in the Discussion, in particular, the GO and KEGG results. This lack of discussion would weaken the conclusions of this manuscript.
Author Response
Dear editor,
I have received the reviewers’ comments on our manuscript entitled “Genome-wide transcriptome analysis reveals that upregulated expression of Aux/IAA genes is associated with defective leaf growth of the slf mutantin eggplant” (Manuscript ID: agronomy-1956893). I am very grateful to the advice from the reviewers and yourself. Several major errors have been corrected, as outlined below. I hope that our paper can now be published in your esteemed journal.
Yours sincerely,
Professor Chen Xueping & Li Na

Reviewer 2 Report
The MS of Du and co-authors is an excellent contribution to crop leaf development, using non-model crops, whose fundamental research is are often negleted. In this work, the authors report on the importance of auxin regulation during leaf development in eggplant. Overall the research approach was well designed, including several complementary methodologies that allowed to unravel the main regulatory mechanism of leaf growth. Despite the high quality of the MS, in my opinion it deserves minor revisions before being published:
(1) Line 29 - replace "cultivated" by "important" (the term cultivated is redundant with crops )
(2) Lines 33 - 34 - ".... but also plays a very important role in human and animal consumption in some poor areas"; I understand the overall idea, but it is not correctly formulated - may be "In eggplant, leaf size has a direct impact on plant growth and yield, compromising in many cases, food security in poor areas"
(3) Line 63 - replace "identified" by "characterized"
(4) Line 67 - delete the conclusive sentence (which is in the abstract and conclusions) - "Our study suggests that auxin is involved in leaf size regulation in eggplant."
(5) Lines 70-72: delete "and the leaf size mutant slf was isolated by EMS from ‘14-345’"; and initiate the paragraph with "In this study, we have used the inbred line ‘14-345’ and its EMS mutant slf, both retrieved from the collection of the State Key Laboratory for Vegetable Germplasm Enhancement and Utilization of Hebei Vegetable Germplasm Re- source Centre."
(6) Line 74 - delete the second "in the greenhouse"
(7) Line 114 - include "program", i.e. "The PCR program used..."
(8) Lines 132-133 - delete the introductory sentence, "We isolated the leaf size mutant slf through an EMS mutated population of eggplant '14-345'"; it is already referred in the M&M
(9) Lines 136-137 - improve "...was tested at the mature stage, where the cell area of slf was 100% larger..."; suggestion: "...was analysed at the mature stage, and the cell area of slf was 100% larger...."
(10) Line 139 - delete "which was".
(11) Line 140 - delete "slf"
(12) Lines 166-167 - "In transcriptome comparison, DEG analysis indicated 1207 DEGs between the first leaf stage of slf and WT (Figure 3A). Further analysis showed that there were 851 upregulated and 356 downregulated DEGs in WT vs. slf leaves." ; I suggest the following change: "Transcriptome analysis identified 1207 Differentially Expressed Genes (DEGs) in first-stage leaves of slf and WT (Figure 3A), 851 of which upregulated and 356 downregulated in WT vs. slf ".
(13) Figure 3A - does it refer to the total number? please clarify the text: do you mean total number of DEGs in slf vs. WT or the other way around?
(14) Line 183 - delete "most"
(15) Line 187 - delete "the" before "genes"
(16) Line 188 - replace "were most significantly enriched in the comparison of slf to WT" by "were significantly enriched in slf in comparison to WT"
(17) Lines 190-191 - same as above (16)
(18) Lines 192-193 - improve the sentence "Thus, the upregulated DEGs were focused on revealing the mechanism of the occurrence of small leaf size in eggplant"; for instance: "Thus, transcriptional activation essentially targeted the mechanism controlling leaf development (size) in eggplant"
(19) Line 195 - delete "genes" after "five" as well as "(Figure 5A&B)"
(20) Lines 197-201 - "Aux1.......in slf" - this is not a result and should be taken from the results and embedded in the discussion
(21) Lines 207-210 - Figure 5 - (A) Is the proposed pathway for the WT? the legend could be improved; (B) these genes are upregulated in slf and downregulated or unchanged in WT; please improve the writing
(22) Line 212: is it possible to include a short explanation on gene selection criteria? Why were DEGs 100111 and 402457 excluded?
(23) Lines 218-221 - "Auxinole......patens" - same as above (20)
(24) Lines 243 - 247 - "Leaf size has considerable economic relevance[61], such as its effects on yield[62, 63] or stress tolerance[64], in crops. Therefore, it is vital to study the mechanisms of leaf size in eggplant for better yield and agricultural management." must be improve, e.g. "Leaf size has considerable economic relevance in crops[61], as it directly influence yield[62, 63] or stress tolerance[64]. Therefore, understanding the mechanisms of leaf development is essential to improve crop management."
(25) Line 250 - delete "another report also showed that Aux/IAAs"
(26) Line 252 - delete "in slf"
(27) Line 254 - delete "is"
(28) Line 265 - the term "no inhibition" must be improved; may be "uncontrolled transcriptional activation....."?
(29) Line 268 - "changed in" - no italic
(30) Lines 270-271 - improve: "..also need further exploration in eggplant"; e.g. "...deserve deeper studies in eggplant"
(31) Lines 278 - "In this study, we isolated a mutant (slf) with a small leaf size by EMS mutagenesis."; this was not exactly the starting point of this work; I suggest ""In this study, we characterized an EMS mutant (slf) with an unusually small leaf size."
Author Response

(The authors gave the same response as above.)

Reviewer 3 Report
Please see my comments in the attached file

Author Response

(The authors gave the same response as above.)

Round 2
Reviewer 1 Report
I appreciate very much the efforts that the authors have devoted to improving their manuscript. I have no more questions.
Author Response
Dear reviewer,
We have received your comments on our manuscript entitled “Genome-wide transcriptome analysis reveals that upregulated expression of Aux/IAA genes is associated with defective leaf growth of the slf mutant in eggplant” (Manuscript ID: agronomy-1956893). We are very grateful that our modifications meet the request of your kindly advices. We hope that you could be reviewer for our next manuscripts and hope this paper can now be published in your esteemed journal.
Yours sincerely,
Professor Chen Xueping & Li Na
State Key Laboratory of North China Crop Improvement and Regulation, Hebei Agricultural University, 071000 Baoding, China.
Tel: +86-18803122833

Reviewer 3 Report
The authors made almost all the changes but did not include the Functional network analysis of upregulated and downregulated genes. I think that analysis will help to understand how other pathways are connected to the Auxin pathway and is not a difficult analysis, so authors must include it.
Author Response
Dear reviewer,
We have received your kindly comments on our manuscript entitled “Genome-wide transcriptome analysis reveals that upregulated expression of Aux/IAA genes is associated with defective leaf growth of the slf mutant in eggplant” (Manuscript ID: agronomy-1956893). We are very grateful to the advice from you. The key question has been answered as below. We hope that our paper can now be accepted in your esteemed journal.
Yours sincerely,
Professor Chen Xueping & Li Na
State Key Laboratory of North China Crop Improvement and Regulation, Hebei Agricultural University, 071000 Baoding, China.
Tel: +86-18803122833
